# Fine cubic Cu$_2$O nanocrystals as highly selective catalyst for propylene epoxidation with molecular oxygen

Wei Xiong[1,7], Xiang-Kui Gu [2,7], Zhenhua Zhang[3], Peng Chai[1], Yijing Zang[4], Zongyou Yu[1], Dan Li[1], Hui Zhang [4], Zhi Liu[4,5] & Weixin Huang [1,6✉]

Propylene epoxidation with O$_2$ to propylene oxide is a very valuable reaction but remains as a long-standing challenge due to unavailable efficient catalysts with high selectivity. Herein, we successfully explore 27 nm-sized cubic Cu$_2$O nanocrystals enclosed with {100} faces and {110} edges as a highly selective catalyst for propylene epoxidation with O$_2$, which acquires propylene oxide selectivity of more than 80% at 90–110 °C. Propylene epoxidation with weakly-adsorbed O$_2$ species at the {110} edge sites exhibits a low barrier and is the dominant reaction occurring at low reaction temperatures, leading to the high propylene oxide selectivity. Such a weakly-adsorbed O$_2$ species is not stable at high reaction temperatures, and the surface lattice oxygen species becomes the active oxygen species to participate in propylene epoxidation to propylene oxide and propylene partial oxidation to acrolein at the {110} edge sites and propylene combustion to CO$_2$ at the {100} face sites, which all exhibit high barriers and result in decreased propylene oxide selectivity.

[1] Hefei National Laboratory for Physical Sciences at the Microscale, Key Laboratory of Surface and Interface Chemistry and Energy Catalysis of Anhui Higher Education Institutes and Department of Chemical Physics, University of Science and Technology of China, 230026 Hefei, People's Republic of China. [2] School of Power and Mechanical Engineering, Wuhan University, 430072 Wuhan, People's Republic of China. [3] Key Laboratory of the Ministry of Education for Advanced Catalysis Materials, Zhejiang Key Laboratory for Reactive Chemistry on Solid Surfaces, Institute of Physical Chemistry, Zhejiang Normal University, 321004 Jinhua, People's Republic of China. [4] State Key Laboratory of Functional Materials for Informatics, Shanghai Institute of Microsystem and Information Technology, Chinese Academy of Sciences, 200050 Shanghai, People's Republic of China. [5] School of Physical Science and Technology, ShanghaiTech University, 201210 Shanghai, People's Republic of China. [6] Dalian National Laboratory for Clean Energy, 116023 Dalian, People's Republic of China. [7]These authors contributed equally: Wei Xiong, Xiang-Kui Gu. ✉email: huangwx@ustc.edu.cn

Propylene oxide (PO) is a platform chemical for numerous commodity chemicals[1], such as polyols and glycol ethers. The current industrial production of PO from propylene involves uses of chlorohydrin or $H_2O_2$ and is cost-ineffective and environment-unfriendly[2,3]. Among various alternative technologies[4–7], propylene epoxidation with $O_2$ to PO is considered most economic and environment-friendly. However, it is meanwhile one of the most challenging catalytic reactions due to unavailable efficient catalysts with high selectivity[8–12]. Cu-based catalysts have been widely studied as a promising catalyst[13–18], but the reported PO selectivity is not satisfying.

Fundamental understanding of active sites for heterogeneous catalytic reactions is an efficient approach to explore novel catalysts. Successful examples have been only a few and they are all based on density functional theory (DFT) calculations[19–23]. Herein, we report a successful exploration of fine cubic $Cu_2O$ nanocrystals (NCs) enclosed with {100} faces and {110} edges as a highly selective catalyst for propylene epoxidation with $O_2$ to PO, guided by an experimental fundamental understanding of the active site. We previously used large rhombic dodecahedral NCs (denoted as d-$Cu_2O$) enclosed with $Cu_2O${110} facets to identify the $Cu_2O${110} facets as the active facet for propylene epoxidation with $O_2$[17], in which, however, reaction temperatures above 150 °C were adopted due to the low density of the active site on the used large d-$Cu_2O$ NCs, favoring the combustion reaction and limiting the acquired PO selectivity. Later large d-$Cu_2O$ NCs with the $Cl^−$ dopant were reported to exhibit enhanced activity in catalyzing propylene epoxidation with $O_2$ and consequently high PO selectivity at low temperatures[9]. Thus, a reasonable strategy to explore highly selective catalysts for propylene epoxidation with $O_2$ to PO is to synthesize uniform fine d-$Cu_2O$ NCs with high densities of $Cu_2O${110} active site, which, unfortunately, has not been realized. Meanwhile, $Cu_2O$ cubes (denoted as c-$Cu_2O$) are enclosed with the {100} faces and {110} edges. We found that the densities of {110} edges are high on $Cu_2O$ cubes (denoted as c-$Cu_2O$) finer than 100 nm and the {110} edge sites on these fine c-$Cu_2O$ NCs, rather than the {100} face sites, are the dominant active site catalyzing the CO oxidation reaction[24]. Intrigued by these findings, we have investigated propylene oxidation with $O_2$ over c-$Cu_2O$ NCs with different sizes and report herein that fine c-$Cu_2O$ NCs with an average size of 27 nm selectively catalyze propylene epoxidation with $O_2$ to PO at temperatures below 110 °C with the $Cu_2O${110} edge sites as the active site. Interestingly, the reaction mechanism for PO production at the $Cu_2O${110} active site was found to switch from weakly adsorbed $O_2$-participating Langmuir–Hinshelwood (LH) mechanism at low temperatures to surface lattice oxygen-participating Mars-van Krevelen (MvK) mechanism at high temperatures.

## Results

**Synthesis and structural characterizations catalysts**. Following previously established procedures[17,25–27], uniform surfactant-free c-$Cu_2O$ NCs with sizes of $27 \pm 4.5$, $106 \pm 12$, and $774 \pm 147$ nm were synthesized (Fig. 1a–c and Supplementary Fig. 1) and denoted as c-$Cu_2O$-27, c-$Cu_2O$-106, and c-$Cu_2O$-774, respectively. BET-specific surface areas of c-$Cu_2O$-27, c-$Cu_2O$-106, and c-$Cu_2O$-774 are 25.5, 11.2, and 1.5 $m^2 g^{-1}$, respectively. XPS spectra (Supplementary Fig. 2) show existences of only adventitious carbon and carbonates on the surfaces of various c-$Cu_2O$ NCs. c-$Cu_2O$ NCs are enclosed with O-terminated $Cu_2O${100} faces and (Cu(I), O)-terminated $Cu_2O${110} edges (Supplementary Fig. 3)[24,28]. Based on the size distributions of various c-$Cu_2O$ NCs, densities of Cu(110) edge sites and their fractions related to total surface Cu sites were estimated to be $6.44 \times 10^{18}$/$g_{Cu2O}$ and 1.61% on c-$Cu_2O$-27, $4.08 \times 10^{17}$/$g_{Cu2O}$, and 0.42% on c-$Cu_2O$-

106, and $7.84 \times 10^{15}$/$g_{Cu2O}$ and 0.06% on c-$Cu_2O$-774 (Supplementary Table 1), respectively. Surface sites of various $Cu_2O$ NCs were probed by CO adsorption at 123 K with in-situ DRIFTS (Supplementary Fig. 4). Vibrational features of adsorbed CO were barely observed for c-$Cu_2O$-774 NCs, but a vibrational feature at 2109 $cm^{-1}$ arising from CO adsorbed at the Cu(I) site[29] emerged for c-$Cu_2O$-106 NCs and grew greatly for c-$Cu_2O$-27 NCs. The Cu(I) sites for CO adsorption on c-$Cu_2O$ NCs exist on the (Cu(I), O)-terminated $Cu_2O${110} edges but not on the O-terminated $Cu_2O${100} faces. Therefore, the density of $Cu_2O${110} edges is too low on large c-$Cu_2O$-774 NCs to be probed by CO adsorption measured with DRIFTS, but becomes high enough on fine c-$Cu_2O$-106 and c-$Cu_2O$-27 NCs.

**Catalytic performance in $C_3H_6$ oxidation with $O_2$**. Various c-$Cu_2O$ NCs exhibit size-dependent catalytic performance in propylene oxidation with $O_2$. As shown in Fig. 2 and Supplementary Fig. 5, c-$Cu_2O$-774 NCs became active at 190 °C and dominantly catalyzed propylene combustion to produce $CO_2$ with $CO_2$ selectivity above 80%. c-$Cu_2O$-27 and c-$Cu_2O$-106 NCs were much more active than c-$Cu_2O$-774 NCs, being catalytically active at 90 °C. Meanwhile, at comparable $C_3H_6$ conversions, c-$Cu_2O$-27 and c-$Cu_2O$-106 NCs exhibited much higher PO selectivities than c-$Cu_2O$-774 NCs. Strikingly, c-$Cu_2O$-27 and c-$Cu_2O$-106 NCs selectively catalyzed the propylene epoxidation with PO selectivity respectively of above 80 and 70% between 90 and 110 °C, but barely catalyzed propylene partial oxidation to acrolein. As the temperature increased above 110 °C, the $CO_2$ selectivity increased rapidly at the expense of PO selectivity, and the acrolein production emerged and grew.

The catalytic performance of c-$Cu_2O$-774 NCs is contributed by the $Cu_2O${100} face sites that selectively catalyze the propylene combustion reaction[17], while the very different catalytic performances of c-$Cu_2O$-27 and c-$Cu_2O$-106 NCs arise from both the $Cu_2O${110} edge sites of enough high densities and the $Cu_2O${100} face sites. The $Cu_2O${110} facets were identified to selectively catalyze the propylene epoxidation reaction[17]. Therefore, the $Cu_2O${110} edge sites of c-$Cu_2O$-27 and c-$Cu_2O$-106 NCs are the dominant surface sites catalyzing propylene oxidation between 90 and 110 °C, giving high PO selectivity, while the contribution from the $Cu_2O${100} face sites increases with the reaction temperature, leading to increased $CO_2$ selectivity at the expense of PO selectivity. These results, on one hand, demonstrate that the $Cu_2O${110} edge sites on c-$Cu_2O$-27 and c-$Cu_2O$-106 NCs are more active than the $Cu_2O${100} face sites, on the other hand, demonstrates that the $Cu_2O${110} site is active in selectively catalyzing the propylene epoxidation with $O_2$ at low temperatures. As far as we know, PO selectivity above 80% in propylene oxidation with $O_2$ catalyzed by c-$Cu_2O$-27 NCs are much higher than all previously reported Cu-based catalysts except the recently-reported $Cl^−$-doped d-$Cu_2O$ NCs[9]. It is noteworthy that the catalytic selectivity of $Cu_2O${110} edge sites of c-$Cu_2O$-27 and c-$Cu_2O$-106 NCs in catalyzing propylene oxidation with $O_2$ between 90 and 110 °C differs very much from that of $Cu_2O${110} face sites of large d-$Cu_2O$ NCs (d-$Cu_2O$-439) which became active only above 150 °C (Supplementary Fig. 6)[17]. In addition to the significantly higher PO selectivity over c-$Cu_2O$-27 and c-$Cu_2O$-106 NCs than over d-$Cu_2O$-439 NCs, acrolein was barely produced for c-$Cu_2O$-27 and c-$Cu_2O$-106 NCs but was a major product for d-$Cu_2O$-439 NCs. Thus the catalytic behavior of $Cu_2O${110} sites in propylene oxidation with $O_2$ sensitively depends on the reaction temperature.

Structures of spent $Cu_2O$ NCs after catalytic performance evaluations at different temperatures were characterized. Both microscopic (Fig. 1d, e and Supplementary Fig. 7) and

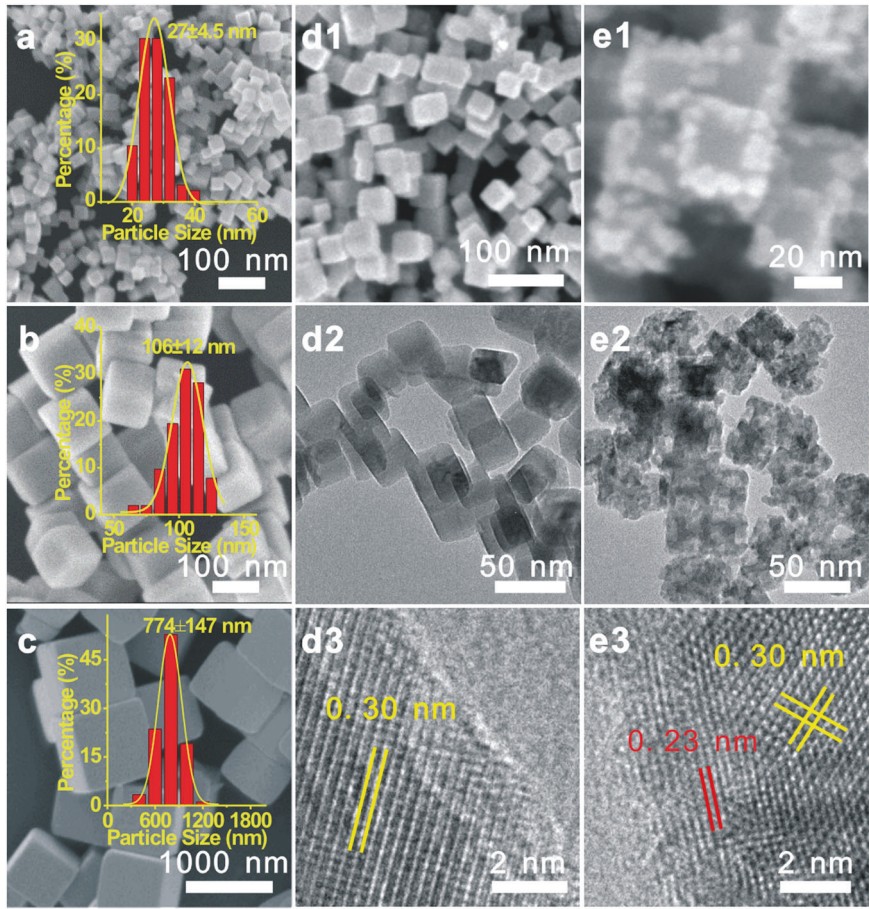

**Fig. 1 Microscopic characterization of Cu₂O NCs.** SEM images and particle size distributions of **a** c-Cu₂O-27, **b** c-Cu₂O-106, and **c** c-Cu₂O-774. SEM, TEM, and HRTEM images of c-Cu₂O-27 after evaluated in C₃H₆ oxidation with O₂ at 90 (**d1**–**d3**) and 150 °C (**e1**–**e3**). Lattice fringes of 0.30 and 0.23 nm correspond to the spacing of the Cu₂O{110} (JCPDS card no. 78-2076) and CuO{111} (JCPDS card no. 89-5899) crystal planes, respectively.

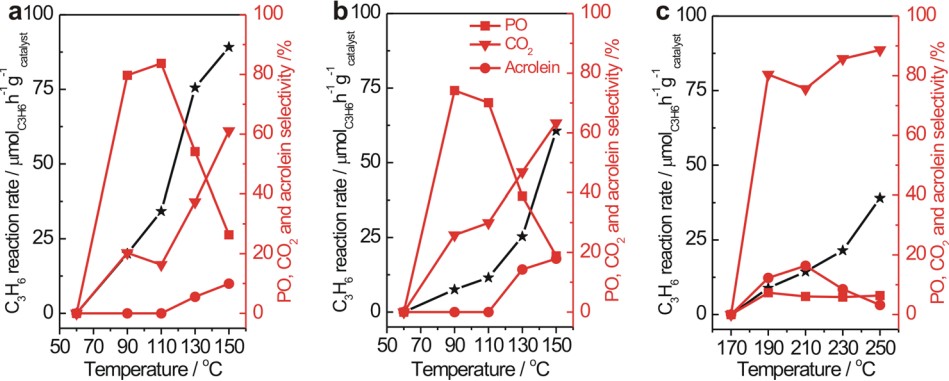

**Fig. 2 Catalytic performance in C₃H₆ oxidation with O₂.** C₃H₆ reaction rate (black) and propylene oxide (PO), acrolein, and CO₂ selectivities (red) of the C₃H₆ oxidation with O₂ catalyzed by **a** c-Cu₂O-27, **b** c-Cu₂O-106, and **c** c-Cu₂O-774 NCs. Reaction condition: 200 mg catalyst, 8% C₃H₆, and 4% O₂ balanced with Ar at a flow rate of 50 mL min⁻¹.

spectroscopic (Supplementary Fig. 8) characterization results show that the structures of spent c-Cu₂O-27 and c-Cu₂O-106 NCs at 90 °C and spent d-Cu₂O-439 at 210 °C are similar to their starting structures, whereas the surfaces of spent c-Cu₂O-27 and c-Cu₂O-106 NCs at 150 °C and spent c-Cu₂O-774 NCs at 210 °C get oxidized, which can be associated with the selective CO₂ production, a highly exothermic reaction. Surface oxidation is more extensive on finer c-Cu₂O-27 NCs than on c-Cu₂O-106 NCs. CuO ad-particles on spent c-Cu₂O-27 and c-Cu₂O-106 NCs

resulting from surface oxidation were observed to locate preferentially at the edges, supporting that the Cu₂O{110} edge sites are where the catalytic reactions dominantly occur. In-situ NAP-XPS spectra of c-Cu₂O-27 NCs under 0.6 mbar C₃H₆ + 0.3 mbar O₂ (Supplementary Fig. 9) do not show obvious surface oxidation at temperatures up to 150 °C. The discrepancy on surface oxidation of c-Cu₂O-27 NCs at 150 °C under catalytic reaction and NAP-XPS measurement conditions can be attributed to the existing pressure gap.

Probed by CO adsorption, the oxidation of $Cu_2O\{110\}$ edges of c-$Cu_2O$-27 and c-$Cu_2O$-106 NCs at 130 and 150 °C also reduces the available surface Cu(I) sites (Supplementary Fig. 10). We found that $C_3H_6$ conversion rates of c-$Cu_2O$-27 and c-$Cu_2O$-106 NCs were proportional to the vibrational peak intensities of CO adsorbed at the surface Cu(I) sites at 90 and 130 °C but not at 150 °C (Supplementary Fig. 11). Therefore, the catalytic performances of c-$Cu_2O$-27 and c-$Cu_2O$-106 NCs up to 130 °C are dominantly contributed by the $Cu_2O\{110\}$ edges with the Cu(I) sites, and the observed decrease of PO selectivity and increase of $CO_2$ selectivity at 130 °C should be due to the more extensive over-oxidation of PO. At 150 °C, although less active than the $Cu_2O\{110\}$ edge sites, the $Cu_2O\{100\}$ face sites of c-$Cu_2O$-27 and c-$Cu_2O$-106 NCs also contribute to the catalytic performance, enhancing the overall $CO_2$ production and selectivity.

Stability of c-$Cu_2O$-27 NCs at 90 °C was further evaluated. $C_3H_6$ conversion kept decreasing with the reaction time, while the PO selectivity gradually increased to almost 100% (Supplementary Fig. 12a). XPS spectra show that the surface of spent c-$Cu_2O$-27 NCs is not oxidized (Supplementary Fig. 12b), while C–H species with the C 1s binding energy at 285.4 eV[12] emerges (Supplementary Fig. 12c). Meanwhile, vibration features of carbonate species (1338 and 1537 $cm^{-1}$)[29], C–O–C (1046 $cm^{-1}$)[30], and C–H (~2928 $cm^{-1}$) groups were observed on the spent catalyst (Supplementary Fig. 12d). These observations indicate that oligomers likely form and accumulate to block the active surface sites on c-$Cu_2O$-27 NCs during the catalytic reaction.

**Reaction mechanism of $C_3H_6$ oxidation with $O_2$.** Figure 3a, b compare $C_3H_6$ and $C_3H_6 + O_2$ temperature-programmed reaction spectra (TPRS) over c-$Cu_2O$-27 and d-$Cu_2O$-439 NCs. Over c-$Cu_2O$-27 NCs (Fig. 3a), PO ($m/z = 58$ and 31) and $CO_2$ ($m/z = 44$) productions did not appear in the $C_3H_6$-TPRS profile but appeared at ~80 °C with similar traces in the $C_3H_6 + O_2$-TPRS profile. Similar acrolein ($m/z = 56$) production traces

appeared at ~100 °C in both $C_3H_6$-TPRS and $C_3H_6 + O_2$-TPRS profiles, and the acrolein production decreased with the temperature increasing in the $C_3H_6$-TPRS profile but increased in the $C_3H_6 + O_2$-TPRS profile. Over d-$Cu_2O$-439 NCs (Fig. 3b), acrolein, PO, and $CO_2$ productions were observed above 200 °C to display similar traces in the $C_3H_6$-TPRS and $C_3H_6 + O_2$-TPRS profiles with more productions in the presence of $O_2$. Thus, no matter at low temperatures over c-$Cu_2O$-27 NCs or at high temperatures over d-$Cu_2O$-439 NCs, the acrolein production by $C_3H_6$ with $O_2$ follows the surface lattice oxygen-participated MvK mechanism, consistent with the previous results[31]. The observed decrease of acrolein with the temperature in the $C_3H_6$-TPRS profile over c-$Cu_2O$-27 NCs likely arises from the insufficient supply of surface lattice oxygen species at the $Cu_2O\{110\}$ edges. The PO production by $C_3H_6$ with $O_2$ at high temperatures over d-$Cu_2O$-439 NCs also follows the MvK mechanism, whereas the PO production at low temperatures over c-$Cu_2O$-27 NCs does not, instead, it should follow a LH mechanism involving surface reactions between co-adsorbed propylene and oxygen species. Therefore, propylene epoxidation with $O_2$ at the $Cu_2O\{110\}$ active site follows the LH mechanism at low temperatures and the MvK mechanism at high temperatures.

In-situ DRIFTS measurements of $C_3H_6$ and $C_3H_6 + O_2$ adsorption on c-$Cu_2O$-27 NCs at different temperatures were carried out to explore the temperature-dependent reaction mechanisms of propylene epoxidation with $O_2$ (Fig. 3c–e). Assignments of observed vibrational features are summarized in Supplementary Table 2. In addition to gaseous $C_3H_6$ (2954 and 1652 $cm^{-1}$), molecularly-adsorbed $C_3H_6$ species at the Cu(I) site of $Cu_2O\{110\}$ edges ($C_3H_6(a)_{Cu}$) (2925 and 1590 $cm^{-1}$) and bridgingly at Cu(I) and O sites of $Cu_2O\{110\}$ edges ($C_3H_6(a)_{Cu,O}$) or bridgingly at O sites of $Cu_2O\{100\}$ face sites ($C_3H_6(a)_{O,O}$) (2925 and 1462 $cm^{-1}$) were observed upon both $C_3H_6$ and $C_3H_6 + O_2$ adsorption at 25 °C, and all vibrational features disappeared upon evacuation, indicating reversible $C_3H_6$ adsorption and absence of $C_3H_6 + O_2$ reaction at 25 °C. At 90 °C, $C_3H_6$

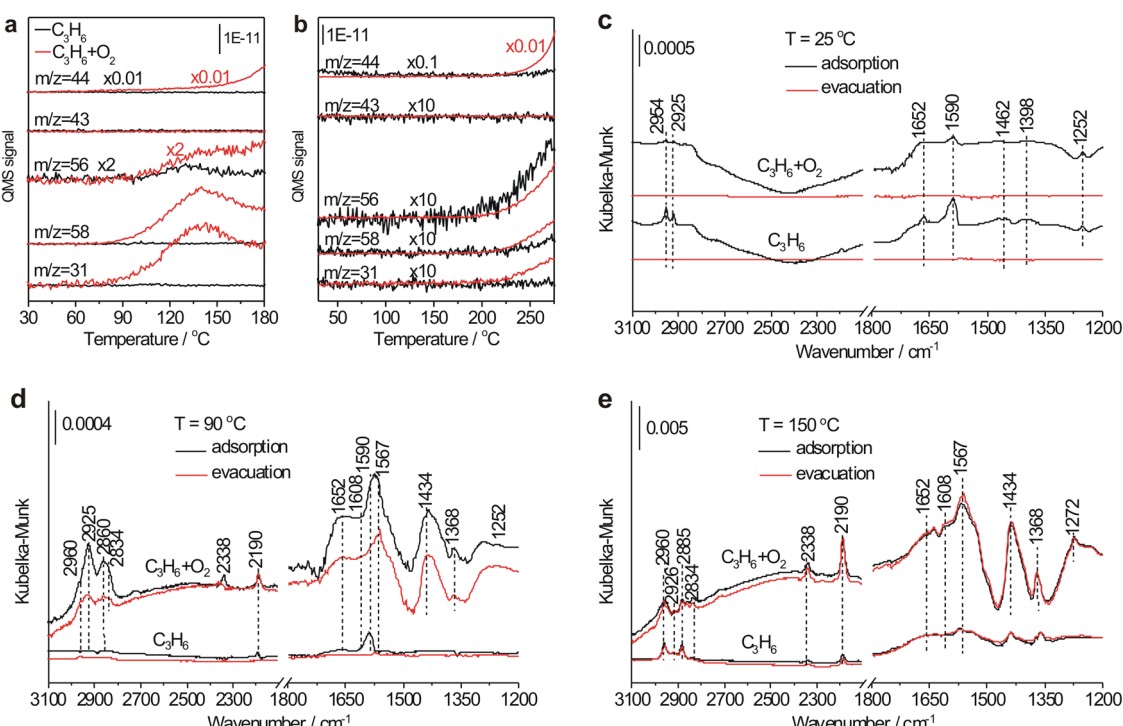

**Fig. 3 Reaction mechanism.** $C_3H_6$-TPRS (8% $C_3H_6$ in Ar) and $C_3H_6 + O_2$-TPRS (8% $C_3H_6 + 4\% O_2$ in Ar) of **a** c-$Cu_2O$-27 and **b** d-$Cu_2O$-439 NCs. DRIFTS spectra of $C_3H_6$ ($P_{C3H6} = 50$ Pa) and $C_3H_6 + O_2$ ($P_{C3H6} = 50$ Pa, $P_{O2} = 25$ Pa) adsorption on c-$Cu_2O$-27 NCs at **c** 25, **d** 90, and **e** 150 °C.

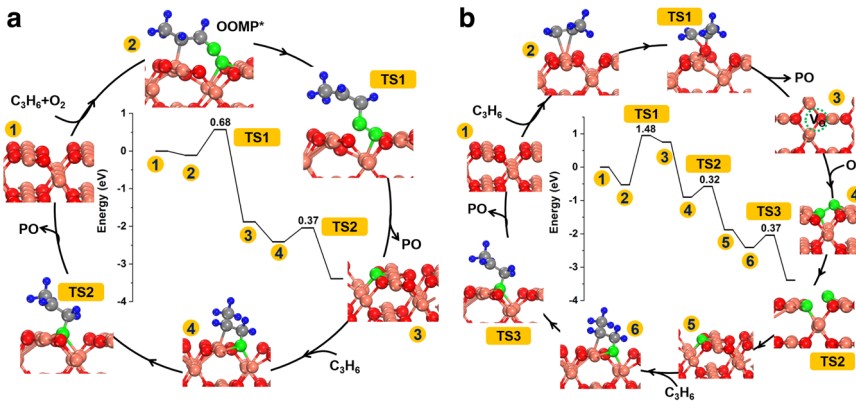

**Fig. 4 DFT calculations.** Energy profile and catalytic cycle along with the optimized structures of intermediates and transition states for propylene epoxidation on $Cu_2O(110)$ via **a** LH and **b** MvK mechanisms. The pink, red, green, gray, and blue spheres represent Cu, O in $Cu_2O$, O in $O_2$, C, and H atoms, respectively.

adsorption dominantly formed $C_3H_6(a)_{Cu}$ species, suggesting that $C_3H_6(a)_{Cu}$ should be more stable than $C_3H_6(a)_{Cu,O}$ and $C_3H_6(a)_{O,O}$, whereas $C_3H_6 + O_2$ adsorption formed not only reversibly-adsorbed $C_3H_6$ species desorbing upon subsequent evacuation but also surface intermediates remaining upon subsequent evacuation, including HCOO(a) (2960, 2885, 1567 and 1368 $cm^{-1}$), allyl adsorbed at the O site $(C_3H_5(a)_O)$ (2834 and 1608 $cm^{-1}$), $CO_2(a)$ (2338 and 2190 $cm^{-1}$), adsorbed acrolein $(C_3H_4O(a))$ (1652 $cm^{-1}$), and allyl adsorbed at the Cu site $(C_3H_5(a)_{Cu})$ (1434 $cm^{-1}$). HCOO(a) and $CO_2(a)$ species are the surface intermediates for the $CO_2$ production, while $C_3H_5(a)$ and $C_3H_4O(a)$ species belong to the surface intermediates for the acrolein production. Previous DFT calculation results suggested that the presence of $O_2$ promoted $C_3H_6$ dehydrogenation reactions to form adsorbed allyl and acrolein species on $Cu_2O$ surfaces[32,33]. Therefore, $C_3H_6 + O_2$ adsorption at 90 °C involves surface reactions between co-adsorbed $C_3H_6(a)$ and oxygen species following the LH mechanism, consistent with the above catalytic performance and TPRS results. Interestingly, few acrolein is produced although $C_3H_5(a)$ and $C_3H_4O(a)$ intermediates are formed, whereas PO is the dominant product but few relevant surface intermediates can be identified. This suggests that the desorption of $C_3H_4O(a)$ to produce gaseous acrolein should exhibit a large barrier and barely occur at 90 °C, whereas surface reactions producing PO can occur. As the temperature increased to 150 °C, the vibrational features of all observed surface intermediates significantly grew, consistent with the enhanced $C_3H_6$ conversion. Meanwhile, $C_3H_6 + O_2$ adsorption gave the same vibrational bands as $C_3H_6$ adsorption but with stronger intensities, supporting the above TPRS result that $Cu_2O\{110\}$-catalyzed propylene oxidation with $O_2$ at high temperatures follows the MvK mechanism. Acrolein production was observed, demonstrating the occurrence of $C_3H_4O(a)$ desorption.

**DFT calculations of $C_3H_6$ oxidation with $O_2$.** DFT calculations were performed to understand the mechanisms of propylene oxidation at the $Cu_2O\{110\}$ active site (Supplementary Table 3). $C_3H_6(a)_{Cu}$ and $C_3H_6(a)_{Cu,O}$ were calculated to exhibit adsorption energy of −0.53 and −0.33 eV and C=C stretch vibrational frequency of 1571 and 1444 $cm^{-1}$, respectively (Supplementary Fig. 13), agreeing with the experimental results. $O_2$ weakly adsorbs with an adsorption energy of −0.25 eV (Supplementary Fig. 14), and its dissociation into two oxygen adatoms exhibits an enthalpy of −0.12 eV but a barrier of 1.16 eV. It can thus be expected that $O_2$ dissociation on the perfect $Cu_2O\{110\}$ surface is

unlikely at low temperatures. Propylene epoxidation with $O_2$ at the $Cu_2O\{110\}$ active site was found to occur via either a LH mechanism or a MvK mechanism (Fig. 4). The LH mechanism initiates via co-adsorption of $C_3H_6$ and $O_2$ to form an oxametallacycle (OOMP) intermediate $(Cu-O-O-CH_2-CH(CH_3)-Cu)$ with a reaction energy of −0.34 eV, which is stronger by 0.26 eV than the formation of $Cu-O-O-(CH_3)-CH-CH_2-Cu$ intermediate. Then the OOMP intermediate dissociates to produce a PO molecule and an atomic O with a barrier of 0.68 eV. Finally, the resulting atomic O readily reacts with $C_3H_6(a)_{Cu}$ to produce another PO molecule with a barrier of 0.37 eV to close the catalytic cycle of propylene epoxidation with $O_2$. Similar mechanisms of $C_3H_6$ epoxidation with molecularly-adsorbed $O_2$ species on IB group metal surfaces were proposed by DFT calculations[34]. The MvK mechanism initiates via the reaction of $C_3H_6(a)_{Cu}$ with surface lattice O to produce a PO molecule and a surface oxygen vacancy $(V_O)$ with a barrier of 1.48 eV. Then $O_2$ readily dissociates at the $V_O$ site to fill it and produce an atomic O with a barrier of 0.32 eV. Finally, the resulting atomic O readily reacts with $C_3H_6(a)_{Cu}$ to produce another PO molecule with a barrier of 0.37 eV to close the catalytic cycle of propylene epoxidation with $O_2$.

Propylene partial oxidation to acrolein at the $Cu_2O\{110\}$ active site following the Mvk mechanism was also calculated (Supplementary Fig. 15). It initiates by an abstract of an α-H atom of $C_3H_6(a)_{Cu}$ to produce $C_3H_5(a)_{Cu}$ or $C_3H_5(a)_O$ with a barrier of 0.75 eV. The $C_3H_5(a)_O$ is more stable than the $C_3H_5(a)_{Cu}$ species by 0.90 eV, resulting in a larger barrier of the $C_3H_5(a)_O$-to-$C_3H_4O(a)$ reaction (1.77 eV) than of the $C_3H_5(a)_{Cu}$-to-$C_3H_4O(a)$ reaction (1.05 eV). Then the resulting $C_3H_4O(a)$ species desorbs to produce an acrolein molecule and create a surface oxygen vacancy with a barrier of 0.96 eV. The H-abstraction reactions of $C_3H_6(a)_{Cu}$ to produce $C_3H_4O(a)$ were previously calculated to be promoted by co-adsorbed $O_2$ species, but the $C_3H_4O(a)$ desorption still exhibited a barrier of 0.99 eV[32]. Meanwhile, the C=C bond breaking of $C_3H_6(a)_{O,O}$ on the $Cu_2O\{100\}$ surface by surface lattice oxygen to eventually produce $CO_2$ was previously calculated with a barrier of 1.05 eV[17].

The above DFT calculation results suggest that the largest barrier is 0.68 eV among elementary surface reactions of LH mechanism for $Cu_2O\{110\}$-catalyzed propylene epoxidation, but is 0.99–1.77 eV of MvK mechanism for $Cu_2O\{110\}$-catalyzed propylene epoxidation, LH and MvK mechanisms for $Cu_2O\{110\}$-catalyzed propylene partial oxidation to acrolein, and MvK mechanism for $Cu_2O\{100\}$-catalyzed propylene combustion. Meanwhile, due to the very small adsorption energy but very large dissociation barrier of $O_2(a)$, $O_2(a)$ is the dominant

adsorbed oxygen species on the stoichiometric $Cu_2O\{110\}$ site and can only form at low temperatures. Thus, at low temperatures at which $O_2(a)$ forms, $Cu_2O\{110\}$-catalyzed propylene epoxidation following the LH mechanism occurs and other reactions with large barriers barely, leading to the high PO selectivity; however, at high temperatures at which $O_2(a)$ is few, $Cu_2O\{110\}$-catalyzed propylene epoxidation following the LH mechanism barely occurs although with low barriers, and $Cu_2O\{110\}$-catalyzed propylene epoxidation and propylene partial oxidation to acrolein and $Cu_2O\{100\}$-catalyzed propylene combustion, all following the MvK mechanism, occur to produce PO, acrolein and $CO_2$, respectively. These DFT calculation results agree well with the experimental observations of temperature-dependent catalytic selectivity of our fine c-$Cu_2O$ NCs in propylene oxidation with $O_2$. Thus, the reactivity and temperature-dependent coverages of various surface species at different surface sites of $Cu_2O$ NCs are responsible for the observed apparent catalytic activity and selectivity in propylene oxidation with $O_2$.

## Discussion

In summary, based on the fundamental understanding of the active site of $Cu_2O$ catalysts for propylene epoxidation with $O_2$, we successfully explore finely-sized cubic $Cu_2O$ NCs with a high density of active $Cu_2O\{110\}$ edge sites as the highly selective catalyst to catalyze propylene epoxidation with $O_2$. Over the c-$Cu_2O$-27 NCs catalyst, $Cu_2O\{110\}$-catalyzed propylene epoxidation with weakly adsorbed $O_2(a)$ species as the active oxygen species exhibits a low barrier and is the dominant reaction occurring at low temperatures, selectively producing PO with a selectivity of above 80%, whereas $Cu_2O\{110\}$-catalyzed propylene partial oxidation and propylene epoxidation and $Cu_2O\{100\}$-catalyzed propylene combustion, all with surface lattice oxygen as the dominant active oxygen species and exhibiting large barriers, occur at high reaction temperatures, producing acrolein, PO and $CO_2$, respectively. These results demonstrate the effectiveness of fundamental understanding in guiding the exploration of efficient catalysts for challenging heterogeneous catalytic reactions.

## Methods

**Chemicals and materials**. All chemical reagents with the analytical grade were purchased from Sinopharm Chemical Reagent Co. $C_3H_6$ (99.95%), $O_2$ (99.999%), CO (99.99%), and Ar (99.999%) were purchased from Nanjing Shangyuan Industrial Factory. All chemicals were used as received.

**Synthesis of c-$Cu_2O$-27 and c-$Cu_2O$-106 NCs**. c-$Cu_2O$-27 and c-$Cu_2O$-106 NCs were synthesized according to the method reported by Chang et al. [26]. To synthesize c-$Cu_2O$-27 NCs, 1 mL $CuSO_4$ aqueous solution (1.2 mol $L^{-1}$) was rapidly injected into 400 mL deionized water at 25 °C. After stirring for 5 min, 1 mL NaOH aqueous solution (4.8 mol $L^{-1}$) was poured into the solution. After stirring for another 5 min, 1 mL ascorbic acid aqueous solution (1.2 mol $L^{-1}$) was injected. Then the solution was kept for another 30 min, and the resulting precipitate was collected by centrifugation, decanting, and washing with distilled water and absolute ethanol, and finally dried in vacuum at RT for 12 h. c-$Cu_2O$-106 NCs were synthesized similarly, except that 0.26 g sodium citrate was added to the initial 400 mL deionized water at 25 °C.

**Synthesis of c-$Cu_2O$-774 NCs**. c-$Cu_2O$-774 NCs were synthesized according to the following typical procedure[25]: 5.0 mL NaOH aqueous solution (2.0 mol $L^{-1}$) was added dropwise into 50 mL $CuCl_2$ aqueous solution (0.01 mol $L^{-1}$) at 60 °C. After adequately stirring for 0.5 h, 5.0 mL ascorbic acid aqueous solution (0.6 mol $L^{-1}$) was added dropwise into the solution. The mixed solution was adequately stirred at 60 °C for 5 h. The resulting precipitate was collected by centrifugation, decanting, and washing with distilled water and absolute ethanol, and finally dried in vacuum at RT for 12 h.

**Synthesis of d-$Cu_2O$-439 NCs**. d-$Cu_2O$-439 NCs capped with oleic acid (OA) (denoted as d-$Cu_2O$-439-OA) were synthesized following Liang et al.'s procedure[27]. Typically, under vigorous stirring, 4 mL OA was mixed with 20 mL of absolute ethanol, and slowly added to 40 mL $CuSO_4$ aqueous solution (0.025 mol $L^{-1}$). The

mixture was heated to 100 °C for 0.5 h. Then 10 mL NaOH aqueous solution (0.8 mol $L^{-1}$) was added. After stirring for another 5 min, 30 mL D-(+)-glucose aqueous solution (0.63 mol $L^{-1}$) was quickly added. The obtained mixture was stirred at 100 °C for another 1 h, and its color changed from black to green, and finally to brick red. The resulting precipitate was collected by centrifugation, decanting, and washing with distilled water and absolute ethanol, and finally dried in vacuum at RT for 12 h.

Capping ligands on as-synthesized d-$Cu_2O$-439-OA NCs were removed following Hua et al.'s procedure[17]. Typically, 150 mg d-$Cu_2O$-439-OA NCs were placed in a U-shaped quartz microreactor and purged in the stream of $C_3H_6 + O_2 + Ar$ gas mixture ($C_3H_6$: $O_2$: Ar = 2: 1: 22) with a flow rate of 50 mL $min^{-1}$ at RT for 0.5 h, and then heated to 215 °C at a heating rate of 5 °C $min^{-1}$ and kept for another 0.5 h. Then the sample was cooled down to room temperature to acquire d-$Cu_2O$-439 NCs.

**In-situ $C_3H_6$ and $C_3H_6 + O_2$ DRIFTS**. Diffuse reflectance infrared spectroscopy (DRIFTS) measurements of chemisorption processes were performed on a Nicolet 6700 FTIR spectrometer equipped with an in-situ DRIFTS reaction cell (Harrick Scientific Products, INC) and a MCT/A detector. 50 mg catalyst was loaded onto the sample stage of the reaction cell. Prior to the experiments, the sample was heated at the desired temperatures at pressures better than 0.1 Pa, and the spectrum was measured and used as the background spectrum, then the adsorbed gas was admitted into the reaction cell to desirable pressures through a leak valve, and the DRIFTS spectra were recorded after the chemisorption processes reached a steady state.

**In-situ $C_3H_6 + O_2$ NAPXPS**. Near-ambient pressure X-ray photoelectron spectroscopy (NAPXPS) measurements were carried out at BL02B01 of Shanghai Synchrotron Radiation Facility[35]. The bending magnet beamline delivered a soft X-ray with photon flux around $1 \times 10^{11}$ photons $s^{-1}$, energy resolution of $E/ \Delta E = 3700$ and beam spot size of ~200 μm × 75 μm on the sample. XPS spectra were calibrated using Au $4f_{7/2}$ binding energy at 84.0 eV. During the NAPXPS experiments, 0.6 mbar $C_3H_6$ and 0.3 mbar $O_2$ were introduced into the chamber, and the c-$Cu_2O$ NCs were heated and stabilized at desirable temperatures for 0.5 h, and then the NAPXPS spectra were measured.

**Structural characterizations**. Power X-ray diffraction (XRD) patterns were conducted on a Philips X'Pert PROS diffractometer using a nickel-filtered Cu Kα (wavelength: 0.15418 nm) radiation source with the operation voltage and operation current being 50 mA and 40 kV, respectively. X-ray photoelectron spectroscopy (XPS) was carried out on an ESCALAB 250 high-performance electron spectrometer using monochromatized Al Kα ($h\nu = 1486.7$ eV) as the excitation source. The likely charging of samples was corrected by setting the binding energy of the adventitious carbon (C 1s) to 284.8 eV. Scanning electron microscope (SEM) images were obtained on a JEOL JSM-6700 field emission scanning electron microscope. Transmission electron microscopy (TEM), high-resolution transmission electron microscopy (HRTEM) were obtained on a JEM-2100F high-resolution transmission electron microscope.

$C_3H_6$-temperature-programmed reaction spectra ($C_3H_6$-TPRS) and $C_3H_6 + O_2$ TPRS were measured in a quartz tube microreactor equipped with an axial quartz sheathed thermocouple and connected to an online mass spectrometer (HIDEN QIC-20). In the $C_3H_6$-TPRS experiments, 50 mg catalyst was pretreated in Ar with a flow rate of 30 mL $min^{-1}$ at 200 °C for 0.5 h and then cooled to 30 °C, then the gas stream was switched to 8% $C_3H_6$ in Ar with a flow rate of 50 mL $min^{-1}$ and the catalyst was heated at a heating rate of 5 °C $min^{-1}$. In the $C_3H_6 + O_2$ TPRS experiments, 50 mg catalyst was pretreated in Ar with a flow rate of 30 mL $min^{-1}$ at 200 °C for 0.5 h and then cooled to 30 °C, then the gas stream was switched to 8% $C_3H_6 + 4\%$ $O_2$ in Ar with a flow rate of 50 mL $min^{-1}$ and the catalyst was heated to the desired temperature at a heating rate of 5 °C $min^{-1}$.

In-situ CO adsorption after catalytic reactions at different temperatures was performed on a Nicolet 6700 FTIR spectrometer equipped with an in-situ low-temperature and vacuum DRIFTS reaction cell (Harrick Scientific Products, Inc.) in order to enhance the chemisorption with minimum interference of gas-phase molecules. The DRIFTS spectra were measured with 256 scans and a resolution of 4 $cm^{-1}$ using a MCT/A detector. 50 mg catalyst was loaded on the sample stage of the reaction cell. Prior to adsorption experiments, the sample was evacuated at 293 K for 1 h at a base pressure of 0.1 Pa and then cooled to 123 K, whose spectrum was taken as the background spectra. Then CO was admitted into the reaction cell to the desirable pressures via a leak valve, and the DRIFTS spectrum was recorded after the chemisorption reached the steady state.

**Catalytic performance evaluation**. Catalytic performance of $Cu_2O$ nanocrystals in propylene oxidation with $O_2$ without any pretreatments was evaluated in a quartz tube microreactor equipped with an axial quartz sheathed thermocouple. 200 mg catalyst was used and heated to the desired reaction temperatures at a rate of 2 °C $min^{-1}$ in a reaction gas mixture ($C_3H_6$: $O_2$: Ar = 2: 1: 22, flow rate: 50 mL $min^{-1}$). After the catalytic reaction reached a steady state, the composition of outlet gas was analyzed using an online Shimazu GC-2014 gas chromatograph equipped with two flame ionization detectors (FIDs) and one thermal conductivity

detector (TCD). One FID was attached to a Stabilwax-DA capillary column (0.53 mm × 60 m) to detect propylene and oxygenates (acetaldehyde, PO, acetone, propionaldehyde, acrolein, acetic acid, and isopropanol) to a detection limit of 1 ppm, and the TCD was attached to a Porapak Q (3 mm × 3 m) and C13x compact column (3 mm × 3 m, Shimazu) to detect $O_2$. A $CH_4$ conversion oven was connected to the end of the TCD to convert trace $CO_2$ to $CH_4$, whose concentration was detected by the other FID. All the lines and valves between the exit of the reactor and the gas chromatographs were heated to 80 °C to prevent condensation of the products. The activity and selectivity of the catalytic reaction were calculated as the following, in which $X_i$ represents conversion, $S_i$ selectivity, $m_i$ mass, and $n_i$ moles of substance $i$, $FC3H6$ represents the flow rate of $C_3H_6$:

$$X_{C3H6} = \frac{n_{oxygenates} + n_{CO_2/3}}{n_{propylene\,in\,feed}} \quad (1)$$

$$S_{PO} = \frac{n_{PO}}{n_{oxygenates} + n_{CO_2/3}} \quad (2)$$

$$S_{Acrolein} = \frac{n_{Acrolein}}{n_{oxygenates} + n_{CO_2/3}} \quad (3)$$

$$S_{CO2} = \frac{n_{CO_2}/3}{n_{oxygenates} + n_{CO_2/3}} \quad (4)$$

$$C3H6\ reaction\ rate = \frac{X_{C3H6} \times F_{C3H6}}{m_{catalyst}} \quad (5)$$

**DFT calculations**. DFT calculations were performed by Vienna ab initio Simulation Package (VASP)[36,37]. The exchange–correlation interaction was described by the Bayesian error estimation functional with van der Waals correlation (BEEF–vdW)[38]. The Kohn–Sham equations were solved by a plane wave basis set with a kinetic energy cutoff of 400 eV. A $Cu_2O(110)$ surface with a (2 × 2) unit cell was modeled by a slab model including four-layer O and seven-layer Cu atoms. To prevent the artificial interaction between the repeated slabs along z-direction, 15 Å vacuum was introduced with correction of the dipole moment. The (2 × 2 × 1) k-point mesh was used to sample the Brillouin zone. During the optimization, the bottom two-layer O and four-layer Cu atoms were fixed in their bulk positions, while the remained atoms and adsorbates were relaxed until the residual forces were less than 0.02 eV Å$^{-1}$. DFT + U correction was used with U-J = 6 eV for Cu 3d-orbitals[39]. Adsorption energies were calculated by $E_{ads} = E_{ad/sub} - E_{ad} - E_{sub}$, where $E_{ad/sub}$, $E_{ad}$, and $E_{sub}$ were the total energies of the optimized adsorbate/substrate system, the adsorbate in the gas phase, and the clean substrate, respectively. Transition states of the elementary steps were located by the climbing-image nudged elastic band (CI-NEB) method[40].

## Data availability

The data supporting the findings of the study are available within the paper and its Supplementary Information. Source data are provided with this paper.

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

## Acknowledgements

This work was financially supported by the National Natural Science Foundation of China (91945301, 21525313, 91745202), the Chinese Academy of Sciences, and the Changjiang Scholars Program of the Ministry of Education of China. The DFT calculations were performed on the supercomputing center of Wuhan University. The NAP-XPS measurements were carried out at the beamline 02B01 of the Shanghai Synchrotron Radiation Facility supported by the National Natural Science Foundation of China under contract no. 11227902.

## Author contributions

W.X carried out the experiments. X.-K.G. carried out the DFT calculations. Z.Z., P.C., Y.Z., Z.Y., D.L., H.Z., and Z.L. assisted with the experiments. W.H. designed and supervised the project. All authors analyzed the data. W.H., W.X., and X.-K.G. prepared the manuscript and other authors commented on the manuscript.

## Competing interests

The authors declare no competing interests.
